# Real-Time Genomic Surveillance during the 2021 Re-Emergence of the Yellow Fever Virus in Rio Grande do Sul State, Brazil

**DOI:** 10.3390/v13101976

**Published:** 2021-10-01

**Authors:** Miguel de S. Andrade, Fabrício S. Campos, Aline A. S. Campos, Filipe V. S. Abreu, Fernando L. Melo, Anaiá da P. Sevá, Jader da C. Cardoso, Edmilson Dos Santos, Lucas C. Born, Cláudia M. D. da Silva, Nicolas F. D. Müller, Cirilo H. de Oliveira, Alex J. J. da Silva, Danilo Simonini-Teixeira, Sofía Bernal-Valle, Maria A. M. M. Mares-Guia, George R. Albuquerque, Alessandro P. M. Romano, Ana C. Franco, Bergmann M. Ribeiro, Paulo M. Roehe, Marco A. B. de Almeida

**Affiliations:** 1Baculovirus Laboratory, Department of Cell Biology, Institute of Biological Sciences, University of Brasilia, Brasília 70910-900, Distrito Federal, Brazil; miguel.souza.andrade@gmail.com (M.d.S.A.); flucasmelo@gmail.com (F.L.M.); bergmann.ribeiro@gmail.com (B.M.R.); 2Bioinformatics and Biotechnology Laboratory, Campus of Gurupi, Federal University of Tocantins, Gurupi 77410-570, Tocantins, Brazil; camposvet@gmail.com; 3State Center of Health Surveillance, Rio Grande do Sul State Health Department, Porto Alegre 90610-000, Rio Grande do Sul, Brazil; aline-campos@saude.rs.gov.br (A.A.S.C.); jaderdacruzcardoso@gmail.com (J.d.C.C.); edmilsondvas@gmail.com (E.d.S.); lucas-born@saude.rs.gov.br (L.C.B.); claudia-dornelles@saude.rs.gov.br (C.M.D.d.S.); 4Insect Behavior Laboratory, Federal Institute of Northern Minas Gerais, Salinas 39560-000, Minas Gerais, Brazil; filipe.vieira@ifnmg.edu.br (F.V.S.A.); cirilohenrique15@gmail.com (C.H.d.O.); alexjunioifnmg@gmail.com (A.J.J.d.S.); 5Department of Agricultural and Environmental Sciences, Santa Cruz State University, Ilhéus 45662-900, Bahia, Brazil; anaiaps@alumni.usp.br (A.d.P.S.); simonini.danilo@gmail.com (D.S.-T.); sofiabv05@gmail.com (S.B.-V.); gralbu@uesc.br (G.R.A.); 6Institute of Basic Health Sciences, Federal University of Rio Grande do Sul, Porto Alegre 90050-170, Rio Grande do Sul, Brazil; nicolas3286@gmail.com (N.F.D.M.); anafranco.ufrgs@gmail.com (A.C.F.); proehe@gmail.com (P.M.R.); 7Flavivirus Laboratory, Instituto Oswaldo Cruz, Fiocruz, Rio de Janeiro 21040-360, Rio de Janeiro, Brazil; angelicamguia@terra.com.br; 8General Coordination of Arbovirus Surveillance, Ministry of Health, Brasília 70058-900, Distrito Federal, Brazil; alessandropecego@gmail.com

**Keywords:** epizootics, phylogenetic analysis, non-human primates

## Abstract

The 2021 re-emergence of yellow fever in non-human primates in the state of Rio Grande do Sul (RS), southernmost Brazil, resulted in the death of many howler monkeys (genus *Alouatta*) and led the state to declare a Public Health Emergency of State Importance, despite no human cases reported. In this study, near-complete genomes of yellow fever virus (YFV) recovered from the outbreak were sequenced and examined aiming at a better understanding of the phylogenetic relationships and the spatio-temporal dynamics of the virus distribution. Our results suggest that the most likely sequence of events involved the reintroduction of YFV from the state of São Paulo to RS through the states of Paraná and Santa Catarina, by the end of 2020. These findings reinforce the role of genomic surveillance in determining the pathways of distribution of the virus and in providing references for the implementation of preventive measures for populations in high risk areas.

## 1. Introduction

Yellow fever (YF) is a viral hemorrhagic fever caused by the Yellow Fever Virus (YFV), the prototype of the genus Flavivirus, family Flaviviridae [1]. In South America, YFV is widely spread and maintained in a sylvatic cycle by transmission between non-human primates (NHP) and blood-feeding mosquitoes mainly from the genus Haemagogus [2].

In Brazil, the area of YFV occurrence extends from the Amazon region, northern Brazil, to the state of Rio Grande do Sul (RS), the southernmost region of the country. The latter is sporadically affected when the virus overflows from the endemic North Region to the Southeast and South Regions in expansion waves with irregular intervals of time [2,3,4,5,6,7]. During such YFV dissemination waves, NHP deaths (hereafter called epizootics) precede human cases, highlighting the importance of the epizootic surveillance system as a tool for early detection of YFV circulation. Prompt action in such episodes allows to actively increase the vaccination coverage of human populations in the vicinity of the epizootic event as well as for mapping virus dispersion [8,9].

Among South American NHP, the genus Alouatta is particularly important for YF surveillance, since its members are most severely affected by YFV [8,10,11,12]. In 2001–2002, following epizootics affecting Alouatta sp., a YFV surveillance program based on monitoring of live and dead NHP was implemented in RS, which has been ongoing since then. Between 2008 and 2009, a new outbreak occurred in RS, causing thousands of NHP deaths and 21 YF human cases. In the subsequent twelve years of continued surveillance, no virus circulation was evidenced in the South Region, which comprises the states of Paraná, Santa Catarina, and RS [3,8,13]. Meanwhile, between 2014 and 2019, a new YFV outbreak was first reported in Tocantins state, in the North Region, from where it spread and reached the Southeast and South Regions, including Brazilian coastal states Espírito Santo, Rio de Janeiro, and São Paulo, causing the largest sylvatic outbreak ever recorded in Brazil [14,15,16,17,18,19,20,21]. Following these events, from 2019 onwards, YFV continued to spread towards the south, arriving in the states of Paraná and Santa Catarina, causing human cases and hundreds of epizootics in NHP [19,22,23].

Altogether, that YF outbreak affected ten states plus the Federal District outside the Amazon region, leaving behind about two thousand human cases and countless NHP deaths. That outbreak was monitored by real-time genomic surveillance [19], and the phylogenetic analysis revealed the existence of at least two main viral sub-lineages occurring in Brazil in that period—the “Yellow Fever Virus Minas Gerais/São Paulo” (hereafter YFV_MG/SP_) and the “Yellow Fever Virus Minas Gerais/Espírito Santo/Rio de Janeiro” (hereafter YFV_MG/ES/RJ_) [4].

Since the virus entered Santa Catarina, RS’s neighboring state, in 2019 [19,24], RS state’s health authorities focused their surveillance efforts at border municipalities. In addition, a surveillance team from the State Health Department was working to promote vaccination and raise awareness of people who live in that region to report NHP deaths [25]. Nonetheless, by the end of 2020 and early 2021, the virus was detected causing deaths of NHP (*Alouatta guariba clamitans*) within RS. In the present study, we report the complete sequence of YFV genomes recovered from NHP in the state of RS and establish its phylogenetic relationship with viral lineages YFV_MG/SP_ and YFV_MG/ES/RJ_ recovered from outbreaks reported in other regions of Brazil.

## 2. Materials and Methods

### 2.1. Ethics Statement

This study comprised analysis of routinely collected surveillance data performed by the state and municipalities health departments and followed the guidelines of the Ministry of Health of Brazil and the Brazilian National Committee for Ethics in Research. All samples were obtained from dead NHP. This study followed the guide to epizootic surveillance in non-human primates and entomology applied to yellow fever surveillance [26] and Institutional Animal Care and Use Committees (IACUCs) review of Nonhuman Primates Research [27]. This study was conducted in accordance with Brazilian legislation under the SISBIO/ICMBio/MMA authorizations for activities with scientific purpose 75734-1 and SISGEN license AF40BCA.

### 2.2. Sample Collection

All NHP samples were collected in municipalities in the Northeast Region of the state of Rio Grande do Sul, Brazil. Samples from liver, kidney, lung, spleen, and heart were collected in the field from dead animals and kept refrigerated (4 °C) or frozen in dry ice (−78 °C) and dispatched to the central office of the State Health Department, where they were stored at −80 °C. All collections followed biosafety protocols and were in accordance with the state YF surveillance strategy carried out by the Environmental Health Surveillance Division, State Center of Health Surveillance, State Health Department of RS, and the Ministry of Health of Brazil [26]. Data concerning the geo-located origin of the animals, date of sampling, and post-mortem findings were recorded.

### 2.3. RT-qPCR

YFV RNAs were extracted from NHP tissues (liver and kidney) samples spotted on Whatman FTA^®^ classic filter paper (GE Healthcare, Chicago, IL, USA). A hole punch was used to excise a single dried 6 mm diameter circle from the dried paper to avoid carryover contamination, the punch was disinfected with bleach, water, and ethanol 100% [28]. The material was then lysed using proteinase K and lysis buffer containing carrier RNA at 56 °C, for 15 min, according to PureLink Viral Mini kit protocol (Invitrogen, Waltham, MA USA). The lysates samples were finally extracted using automated extraction (Loccus, Extracta Kit FAST, Cotia, SP, Brazil). Viral RNA was detected using two previously published RT-qPCR protocols [29]. Positive samples at RT-qPCR, with cycle thresholds (CT) below 25, were sent for sequencing in the Baculovirus Laboratory, at the University of Brasilia.

### 2.4. Genome Sequencing

All samples that met the previous criteria were submitted to cDNA synthesis protocol using LunaScript™ RT SuperMix Kit (NEB, Ipswich, MA, USA) following the manufacturer’s instructions. Then, a multiplex tiling PCR was performed using the previously published YFV primers [30] and 40 cycles (denaturation: 95 °C/15 s and annealing/extension: 65 °C/5 min) of PCR using Q5 high-fidelity DNA polymerase (NEB, Ipswich, MA, USA). Amplicons were purified using 1× AMPure XP beads (Beckman Coulter, Indianapolis, IN, USA) and cleaned-up PCR product concentrations were measured using a QuantiFluor^®^ dsDNA System assay kit on a Quantus™ Fluorometer (Promega, Madison, WI, USA). DNA library preparation was performed using the Ligation sequencing kit SQK-LSK309 (Oxford Nanopore Technologies, Oxford, United Kingdom) and the Native barcoding kit (EXP-NBD104 and EXP-NBD114; Oxford Nanopore Technologies, Oxford, UK). The sequencing library (22 samples and a negative control per run) was loaded onto a R9.4 flow cell (Oxford Nanopore Technologies, Oxford, United Kingdom) and sequenced between 6 to 18 h using MiNKNOW software (Oxford Nanopore Technologies, Oxford, United Kingdom). The RAMPART (Version 1.2.0, ARTIC Network, Oxford, United Kingdom) package was used to monitor coverage depth and genome completion. The resulting Fast5 files were basecalled and demultiplexed using Guppy (Version 4.4.2, Oxford Nanopore Technologies, Oxford, United Kingdom). Variant calling and consensus genome assembly were carried out with Medaka (Version 1.0.3, Oxford Nanopore Technologies, Oxford, United Kingdom) using the sequence JF912190 as the reference genome.

### 2.5. Phylogenetic Analysis

To perform phylogenetic analysis, we selected from NCBI all near-complete YFV sequences (YFV-set-1, *n* = 359, sequences > 8 kb excluding sequences from vaccine and patents). A subset of sequences belonging to recent (2015 to 2021) extra-Amazonian region waves, including clades YFV_MG/ES/RJ_ and YFV_MG/SP_ (YFV-subset, *n* = 264), were selected. Metadata such as sample collection date and geographic coordinates were retrieved from GenBank files or gathered manually from genome associated publications. The genomes recovered here (*n* = 22) combined with YFV-set-1 were aligned with MAFFT v.7.480 [31]. The Maximum-likelihood tree was inferred using IQTREE, with the GTR+F+I+Γ4 model. YFV-subsets combined with the newly determined genomes were used to construct a time-scaled tree in Nextstrain (https://nextstrain.org/ncov, accessed on 23 August 2021). The new genome sequences were sent to the NCBI GenBank database under accession numbers MZ712127 to MZ712149.

### 2.6. Epidemiological and Geographic Information

Epidemiological data on human cases (from 2016 to March 8th, 2021) and epizootics (from 2014 to March 8th, 2021) of YF in Brazil were provided by the Ministry of Health of Brazil [32]. The total cases in NHP of previous epizootics in RS were provided by the Division of Environmental Health Surveillance from Rio Grande do Sul State Health Department. Maps presenting the results were generated through free software QGIS version 2.18 (Maisons-Laffitte, France) and GeoPandas (Python, NumFOCUS, Austin, TX, USA).

## 3. Results

Between January and 08 March 2021 (date of collection of the last sample whose sequence was recovered here), 78 epizootics were notified to the RS State Health Department and Ministry of Health of Brazil surveillance, with sample collection performed in 42 cases (54%). From these, 34 (81%) tested positive for YFV by RT-qPCR assay. All confirmed YFV-positive NHP were *Alouatta guariba clamitans* and it became evident that, at the date of collection of the last sample sequenced, the virus remained in circulation in the state for at least 10 weeks, with epizootics occurring mainly in the Northeast Region of RS, near to the border with Santa Catarina state (Figure 1).

Twenty-two near-complete YFV genomes were generated from liver samples from dead NHP collected in eight municipalities of RS. Sequenced samples displayed median Ct values of 12 (range 8–20) at RT-qPCR (Table 1). Phylogenetic analyses revealed that the sequences generated here clustered within the South America I clade [33,34] (Figure 2) and did not group with genomes recovered from previous outbreaks that occurred in RS in 2001 and 2008 (GenBank accession numbers JF912189 and KY861728, respectively) [35,36]. These findings indicate that YFV associated with the epizootics reported here were consequent to a new introduction of virus from other states and are not related to possible re-emergence of previously circulating YFV within RS.

To examine the spatial and evolutionary dynamics of the dispersion of YFV in the epizootic reported here, we made an analysis using a subset of sequences belonging to the South America 1 genotype on Nextstrain. The time-scaled phylogenetic tree (Figure 3A) shows that the genomes of YFV from RS in 2021 clustered with the YFV_MG/SP_ sub-lineage, named Yellow Fever Virus Minas Gerais/São Paulo/Rio Grande do Sul (hereafter YFV_MG/SP/RS_), revealing that the origin of these isolates is São Paulo. Furthermore, the time-scaled phylogenetic tree (Figure 3A) suggests more than one entry in RS. Despite the lack of genomic data from Paraná and Santa Catarina, epidemiological data of epizootics and human cases due to YF from São Paulo and South Region (Paraná, Santa Catarina and RS), as shown in Figure 3B, suggest that the recent YFV dispersion wave achieved RS through Paraná (2018/2021) and Santa Catarina (2019/2021).

## 4. Discussion

In this study, we applied real-time genomic surveillance to generate sequences of YFV from NHP found dead by the YF surveillance system in RS state, southern Brazil, at the early stages of the re-introduction of the virus in 2021. Such re-introduction led the state to declare a Public Health Emergency of State Importance. To examine the distribution, the spread, and lineage of the YFV virus involved in the current epizootics, 22 complete to nearly complete YFV genomes were recovered from NHP samples collected in the northeast of RS state during the first peak of the epizootic (early 2021) and sequenced.

In Brazil, over the past six years, previous genetic analyses have indicated the circulation of at least two distinct sub-lineages, which spread in two different transmission routes. The first one was YFV_MG/ES/RJ_, which circulated in the east of Minas Gerais in 2016, entering the west of Espírito Santo and following to the north of the state of Rio de Janeiro. From there, in 2017, the virus spread southwards, arriving at the border with São Paulo in 2018 [4,15,30,37,38]. Approximately in March 2017, NHP samples infected by this sub-lineage were also found in Bahia, a Northeast Region state in Brazil [39]. Meanwhile, the second sub-lineage, named YFV_MG/SP_, circulated in the west of Minas Gerais and the northwest of São Paulo in 2016–2017, from where it advanced towards the south and east of that state, reaching the most densely populated region of the country between 2017 and 2018 [10,40]. Between 2018 and 2021, YFV was detected in NHP and humans in the states of Paraná and Santa Catarina, revealing its movement southwards (although no genome sequences of the viruses circulating in those states were available at the time of writing this report) [19,22,23]. Despite the lack of genomic information available from Paraná and Santa Catarina, the spatio-temporal distribution of the NHP epizootic and the phylogenetic analyses revealed that the genomes of viruses circulating in RS in 2021 clustered along with sub-lineage YFV_MG/SP_ (now called YFV_MG/SP/RS_). These findings suggest that this sub-lineage had moved through the three states of the South Region of Brazil.

Previous records of occurrence of YF in RS showed that the northwestern border of RS with Argentina has historically been the first affected area for YF cases in NHP, as happened in 1947, 1966, 2001, 2002, and 2008/2009 [3,8,35,36,41,42]. In the epizootic here reported, the virus spread through a different route, from São Paulo to Paraná and Santa Catarina, landing in RS through its Northeast region. Sequencing samples from these states are essential to better understand the dispersion pattern of YFV while circulating in the South Brazil Region.

Interestingly, the sequences obtained here represent virus samples of the farthest place reached by YFV dispersion in Brazil, since the spread from the endemic area, approximately in 2014, in this expansion wave. Rio Grande do Sul is the southernmost limit for NHP distribution in the Americas [43] and the farthest state from the YF endemic area, Amazon region, which is the source of YFV in the events of viral expansion [2]. Consequently, it is expected to be the southernmost limit for YFV spread in the Americas too. Possibly, the 2021 YFV lineage from RS presents accumulated genetic changes over seven years of circulation, since the first detection in an extra endemic area, in 2014, in the State of Tocantins [19]. A total of 69 SNVs, 50 synonymous and 19 nonsynonymous, were detected (Appendix A) when we compared our 22 genomes with genome MK3233804, collected in the state of Goiás on 17 August 2015, the oldest genome of this current extra endemic wave. Furthermore, two unique non-synonymous mutations are present in all RS samples, one in the capsid region and the other in NS4A (Appendix A) and possibly, they may have been fixed at some point during the virus spread through SP/PR/SC. This type of analysis is limited due to the absence of YFV genomes from PR and SC, and future studies should be carried out.

Several environmental and ecological issues must be addressed to better understand the dynamics of virus dispersion. Epidemic outbreaks in the sylvatic cycle of YFV demand a high density of competent vectors and the presence of susceptible NHP, acting as amplifiers for the virus [11,14,36,44]. This probably interferes with the speed and even the direction of viral spread. Noteworthy, the YFV expansion wave (2014–2021) revealed the suitability of climate and ecological conditions for the occurrence of YFV outbreaks in several Brazilian regions [4,45,46,47]. Such spreading events included even the three southernmost states of Brazil (Paraná, Santa Catarina, and Rio Grande do Sul), with a subtropical temperate climate and great losses of forest areas, through which viral circulation could more easily occur [48]. Moreover, particularly in RS, only two genera of NHP, Alouatta and Sapajus, have been detected [49], limiting the amplification through susceptible NHP hosts to these species.

In the present report, we examine the likely origin of YFV strains that were recovered from NHP in RS, the southernmost state of Brazil, in the current YFV wave which was found to be heading south within the country, as opposed to previous outbreaks in the state that had been introduced via Argentinian border. Phylogenetic analyses revealed that such sequences are related to the lineage that occurred in São Paulo during YFV circulation in 2017–2018. However, further studies of the virus dispersion including ecological, climatic, and anthropogenic factors associated with the disease cases are needed to improve the predictive power, allowing rapid decisions into surveillance and prevention efforts. Additional studies shall be conducted to compare eventually available YFV genomic sequences from specimens collected in the states of Paraná and Santa Catarina to provide further evidence of the dissemination of the virus within the country. Real-time genomic surveillance must be continued in order to allow deeper understanding of the factors driving the transmission and dispersal of the Yellow Fever Virus in NHP.

In Brazil, YF vaccine is freely offered by the Brazilian Unified National Health System (SUS, in Portuguese), as part of routine immunization, but the adherence in the target population (from 9 months to 59 years of age) is unequal, mainly when not accompanied by special communication of risk to society and health professionals, which leads to a very heterogeneous vaccine coverage (VC) across the country [19]. The area with vaccine recommendation was expanded in the country, from 1,300 municipalities and 31.3 million inhabitants (19.3% of the Brazilian population) in 1998 to the entire Brazilian territory in 2020, with an estimated population of 211.5 million inhabitants, aiming to reach the VC goal, which is 95% of the target population [50]. Based on 2019 data, global VC in Rio Grande do Sul (67%) is lower than in São Paulo (77%) and Minas Gerais (95%) States. After the past outbreaks occurred in São Paulo and Minas Gerais, which generated hundreds of cases, a huge intensification of vaccination was triggered in those states. It is also important to note that although the global VC of a state reaches a satisfactory percentage, it is usually not homogeneous for all municipalities. For instance, from the 497 municipalities in RS, 175 (35%) have VC equal to or greater than 95%, 245 (49%) have VC between 50% and 94%, and 77 (16%) have VC below 50%. In our study area, due to a previous YFV circulation (in 2009) and also a vaccination intensification that took place in 2019/2020, the municipalities hit by YFV at the beginning of this viral dispersion (from January 2021) had a high VC. It led us to believe that this area was not a candidate for human cases occurrence, and, actually, to date (September 2021), no human cases were recorded in RS during the current outbreak, despite the wide YFV circulation among NHP.

This report highlights the importance of capacity building for inter-institutional exchange of data and human resources, to strengthen epidemic surveillance and outbreak management during pandemics. Additionally, our study represents a model for YFV response that reinforces the need to bring together epidemiology, surveillance, vaccination, and genomic tracking.

## Figures and Tables

**Figure 1 viruses-13-01976-f001:**
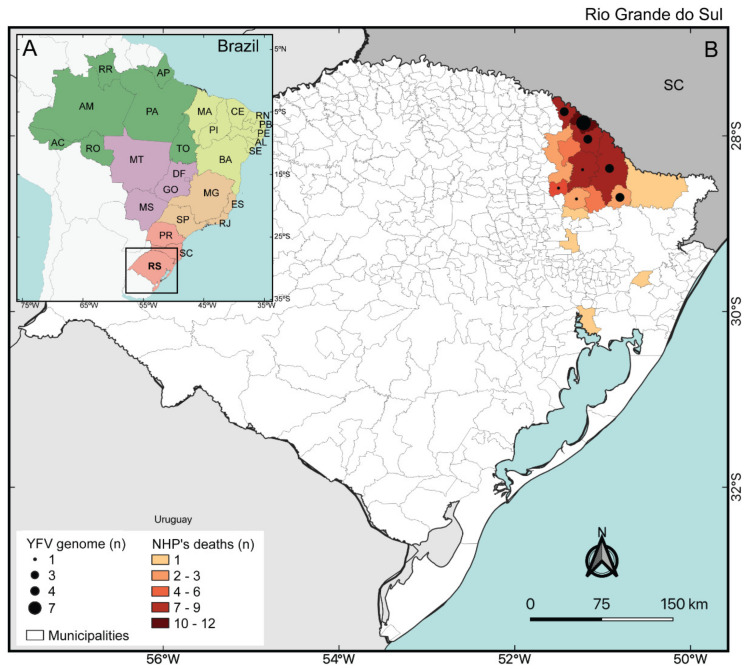
(**A**) Brazilian regions: North (green); Northeast (yellow); Central-West (purple); Southeast (orange); South (red). (**B**) Geographical distribution of YFV NHP cases in Rio Grande do Sul. The municipalities with NHP deaths positive at RT-qPCR are highlighted. Numbers of genomes recovered per municipality are illustrated by black dots.

**Figure 2 viruses-13-01976-f002:**
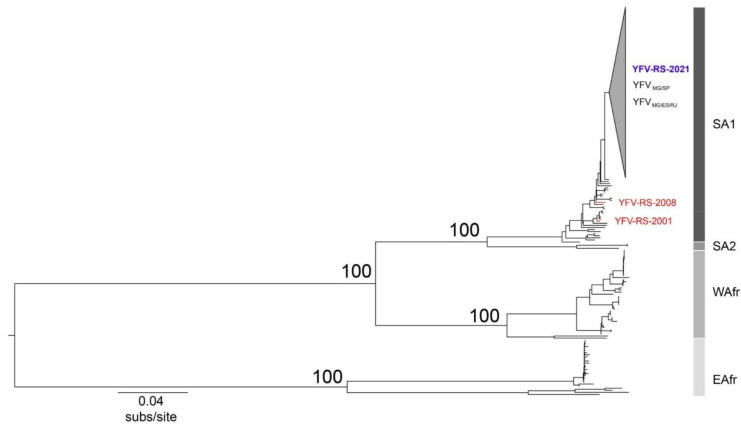
Phylogenetic tree of YFV based on 381 near-complete genomes. The gray collapsed group includes YFV_MG/ES/RJ_ and YFV_MG/SP_ clades and all genomes from RS sequenced in this study. South America I, South America 2, West Africa, and East Africa genotypes are indicated. Genomes recovered from previous epizootics in RS named YFV-RS-2001 (JF912189) and YFV-RS-2008 (KY861728) are highlighted in red.

**Figure 3 viruses-13-01976-f003:**
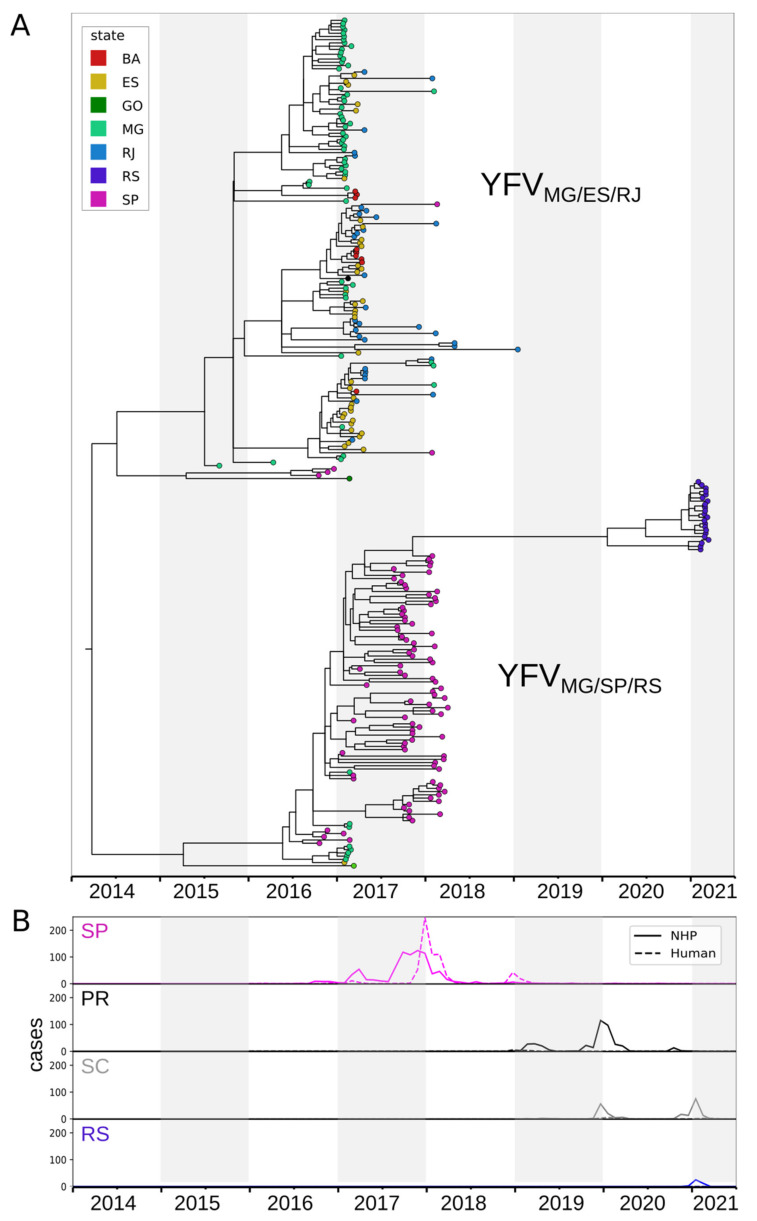
Spatio-temporal YFV spread. (**A**) Time-scaled phylogenetic tree of YFV_MG/ES/RJ_ and YFV_MG/SP/RS_ sub-lineages. (**B**) Epizootics and human cases of YF reported in São Paulo (SP), Paraná (PR), Santa Catarina (SC), and Rio Grande do Sul (RS) by month (Source: Ministry of Health of Brazil).

**Table 1 viruses-13-01976-t001:** List of genomes generated in this study (*n* = 22) showing date of collection, sample name, GenBank accession code, latitude, longitude, cycle threshold (Ct), and coverage. All samples were collected in 2021.

Date of Collection.	Sample Name	Accession	Lat	Long	Ct	Coverage
4 March	André da Rocha 02	MZ712127	−28.5850	−51.5757	20	1008
22 February	Barracão 02	MZ712128	−27.7315	−51.3688	11	247
22 February	Barracão 03	MZ712129	−27.7315	−51.3688	9	153
22 February	Barracão 04	MZ712130	−27.7315	−51.3688	15	666
19 February	Esmeralda 01	MZ712131	−28.0930	−51.1124	12	360
24 February	Esmeralda 02	MZ712132	−28.1698	−50.9228	15	207
25 February	Esmeralda 03	MZ712133	−27.9754	−51.0557	9	153
25 February	Esmeralda 04	MZ712134	−27.9754	−51.0557	11	294
8 March	Ipê 01	MZ712135	−28.7591	−51.2527	12	2024
8 February	Monte Alegre dos Campos 01	MZ712136	−28.5355	−51.5023	10	30
26 February	Monte Alegre dos Campos 02	MZ712137	−28.7591	−51.2527	11	84
3 March	Monte Alegre dos Campos 03	MZ712138	−28.6724	−50.7997	14	722
20 February	Muitos Capões 01	MZ712139	−28.2196	−51.2135	8	309
25 January	Pinhal da Serra 01	MZ712140	−27.8757	−51.2260	20	194
3 February	Pinhal da Serra 02	MZ712141	−27.8757	−51.2260	11	156
3 February	Pinhal da Serra 03	MZ712142	−27.8757	−51.2260	11	135
8 February	Pinhal da Serra 05	MZ712143	−27.8346	−51.1995	10	284
11 February	Pinhal da Serra 07	MZ712144	−27.8843	−51.1632	13	53
19 February	Pinhal da Serra 09	MZ712146	−27.8403	−51.2505	10	1572
22 February	Vacaria 01	MZ712147	−28.2900	−50.8116	9	568
22 February	Vacaria 02	MZ712148	−28.2900	−50.8116	10	416
24 February	Vacaria 04	MZ712149	−27.9216	−51.2187	11	533

## Data Availability

Not applicable.

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
