# Peer review of "Real-Time Genomic Surveillance during the 2021 Re-Emergence of the Yellow Fever Virus in Rio Grande do Sul State, Brazil"

_viruses, 2021, doi:10.3390/v13101976_

Round 1

Reviewer 1 Report

Andrade et al performed genomic surveillance and  phylogenetic analysis on epizootic strains of yellow fever virus in Brazil’s southern-most state; Rio Grande do Sul. The work presented is quite valuable, as it illustrates which South American sub-lineages of YFV are still actively circulating to date after the larger outbreaks taking place in 2017. Given the low rate of YFV vaccine coverage in Southern Brazil as compared to the North, as well as the existence of susceptible non-human primates in the area, this surveillance work is critical to monitoring for potential emergence of YFV into urban cycles. Therefore, the work is appropriate for publication, albeit that there are some concerns with the present study that require clarification and /or revision prior to publication.

Major Concerns

  1. The authors note that the epizootics surveyed are not associated with human cases. What is critical to note here is the relative vaccine coverage in the state as compared to other states such as Minas Gerais and Sao Paulo. Without the context of vaccine coverage, an unfamiliar reader has no understanding whether these epizootics failed to emerge into an urban cycle because of a limited scale, or whether vaccine coverage prevented entry into the human population. This point should be addressed in the discussion

  1. While the phylogenetic data undoubtedly valuable as is, I am slightly discouraged by the lack of mosquito surveillance data. There would be incredible value for example to have a mosquito sampling aspect of the study conducted at sites where dead Alouatta were found. Presently the study is solely limited to strains that caused lethal disease in Alouatta, but may be neglecting circulating strains that are not efficiently making the jump from mosquito to monkey. This is particularly important given that there is potential for an independent enzootic cycle to be established in Rio Grande do Sul, removing the need for the virus to “migrate” from the North.                                        Since the surveillance has already taken place, and I feel it is unreasonable to ask for accompanying mosquito data that may not exist, I do feel that a discussion of the potential for long term enzootic circulation is warranted. This particularly important given the paper's overall message of the value for genomic surveillance strategies.

Reviewer 2 Report

This study reports genomic sequences and establishes phylogenetic relationships of YFV recovered from non-human primates in Rio Grande do Sul, Brazil.  The study design is straightforward and the data support the conclusions that are drawn.  The manuscript is also very well written. I only have two suggestions to polish the manuscript:

  1. In addition to the phylogenetics, it might be interesting to discuss the sequence information in a bit more granular fashion.  Were there interesting SNPs, etc. in key proteins or regulatory regions that might contribute to virulence/pathogenesis?
  2. The ‘date accessed’ information should be removed from references in the list at the end of the paper as it is not pertinent to the report.
